# Evaluation of the Small Heat Shock Protein Family Members *HSPB2* and *HSPB3* in Bladder Cancer Prognosis and Progression

**DOI:** 10.3390/ijms24032609

**Published:** 2023-01-30

**Authors:** Despoina D. Gianniou, Aimilia D. Sklirou, Maria-Alexandra Papadimitriou, Katerina-Marina Pilala, Konstantinos Stravodimos, Margaritis Avgeris, Andreas Scorilas, Ioannis P. Trougakos

**Affiliations:** 1Department of Cell Biology and Biophysics, Faculty of Biology, National and Kapodistrian University of Athens, 15784 Athens, Greece; 2Department of Biochemistry and Molecular Biology, Faculty of Biology, National and Kapodistrian University of Athens, 15701 Athens, Greece; 3First Department of Urology, “Laiko” General Hospital, School of Medicine, National and Kapodistrian University of Athens, 11527 Athens, Greece; 4Laboratory of Clinical Biochemistry-Molecular Diagnostics, Second Department of Pediatrics, School of Medicine, National and Kapodistrian University of Athens, “P. & A. Kyriakou” Children’s Hospital, 11527 Athens, Greece

**Keywords:** bladder cancer, biomarker, chemotherapy, *HSPB2*, *HSPB3*, molecular chaperone, prognosis, recurrence

## Abstract

Bladder cancer (BlCa) represents the sixth most commonly diagnosed type of male malignancy. Due to the clinical heterogeneity of BlCa, novel markers would optimize treatment efficacy and improve prognosis. The small heat shock proteins (sHSP) family is one of the major groups of molecular chaperones responsible for the maintenance of proteome functionality and stability. However, the role of sHSPs in BlCa remains largely unknown. The present study aimed to examine the association between *HSPB2* and *HSPB3* expression and BlCa progression in patients, and to investigate their role in BlCa cells. For this purpose, a series of experiments including reverse transcription-quantitative PCR, Western blotting, MTT assay and flow cytometry were performed. Initial analyses revealed increased vs. human transitional carcinoma cells, expression levels of the *HSPB2* and *HSPB3* genes and proteins in high grade BlCa cell lines. Therefore, we then evaluated the clinical significance of the *HSPB2* and *HSPB3* genes expression levels in bladder tumor samples and matched adjusted normal bladder specimens. Total RNA from 100 bladder tumor samples and 49 paired non-cancerous bladder specimens were isolated, and an accurate SYBR-Green based real-time quantitative polymerase chain reaction (qPCR) protocol was developed to quantify *HSPB2* and *HSPB3* mRNA levels in the two cohorts of specimens. A significant downregulation of the *HSPB2* and *HSPB3* genes expression was observed in bladder tumors as compared to matched normal urothelium; yet, increased *HSPB2* and *HSPB3* levels were noted in muscle-invasive (T2–T4) vs. superficial tumors (TaT1), as well as in high-grade vs. low-grade tumors. Survival analyses highlighted the significantly higher risk for post-treatment disease relapse in TaT1 patients poorly expressing *HSPB2* and *HSPB3* genes; this effect tended to be inverted in advanced disease stages (muscle-invasive tumors) indicating the biphasic impact of *HSPB2*, *HSPB3* genes in BlCa progression. The pro-survival role of *HSPB2* and *HSPB3* in advanced tumor cells was also evident by our finding that *HSPB2*, *HSPB3* genes expression silencing in high grade BlCa cells enhanced doxorubicin toxicity. These findings indicate that the *HSPB2*, *HSPB3* chaperone genes have a likely pro-survival role in advanced BlCa; thus, they can be targeted as novel molecular markers to optimize treatment efficacy in BlCa and to limit unnecessary interventions.

## 1. Introduction

Bladder cancer (BlCa) ranks as the sixth most common male malignancy worldwide and the second more frequent type of urological cancer in men, being associated with high morbidity and mortality [1,2,3]. The majority of bladder tumors originate from the epithelium (urothelium) that covers the inner surface of bladder and are classified as either non-muscle invasive bladder cancer (NMIBC) (Tis, Ta, T1), or muscle invasive bladder cancer (MIBC) (T2–T4). Infiltration of the muscular layer constitutes tumor depth of invasion that is the main clinicopathological factor which determines disease staging, prognosis and therapeutic approach [4]. Approximately 75% of the patients are diagnosed with NMIBC and 50–70% of them will relapse following treatment, while the other 25% display muscle-invasive or metastatic BlCa [5,6].

Bladder cancer treatment varies by stage, with TaT1 patients being treated with transurethral resection of the bladder tumor (TURBT); which may be followed by intravesical Bacillus Calmette-Guérin (BCG) immunotherapy or chemotherapy, while the standard treatment for MIBC is radical cystectomy (RC) [7]. Nonetheless, the prediction of therapy response and disease outcome remains challenging, due to the high clinical heterogeneity of bladder tumors, resulting to lifelong surveillance strategies. At present, cystoscopy is the golden rule for both disease diagnosis and monitoring, however, it is highly invasive and incurs significantly healthcare systems costs [8,9]. Thus, the identification of novel molecular disease markers could significantly ameliorate patients’ discomfort, supporting personalized treatment decisions.

The family of small heat shock proteins (sHSPs) in humans includes ten molecular chaperones (HSPB1-HSPB10) [10] being involved in maintaining proteome stability and functionality [11,12,13]. sHSPs can also play an anti-apoptotic role by modulating molecules such as JNK, AKT and NF-κB [14]. It is thus not surprising that they have been functionally implicated in many diseases, including cancer as they regulate cell proliferation, differentiation, invasion, metastasis, death and tumor cells’ recognition by the immune system [15,16]. For instance, HSPB1, one of the most studied members of sHSPs, has been found to be overexpressed and plays an important role in a variety of human cancers including breast [17], colon [18], liver [19] and bladder [20]. HSPB1, thus, has already been examined as a potential therapeutic target in BlCa. Specifically, OGX-427 (Apatorsen, OncoGenex, Vancouver, BC, Canada), a sequence of second-generation antisense oligonucleotides (ASO) generated using a 2′-O-(2-methoxy) ethyl (2′-MOE) backbone, which targets *HSPB1* mRNA, is reported to enhance chemosensitivity of cancer cells to therapeutic agents; to exert anti-proliferative effects and to also inhibit tumor growth in mice [20,21]. Additionally, OGX-427 has been studied in many malignancies, including lung (NCT01829113), pancreatic (NCT01844817), prostate (NCT01120470, NCT01681433, NCT00487786) and bladder cancer (NCT01454089, NCT01780545).

Despite the fact that HSPB2 was first discovered in 1997 [22], its mechanistic details remain unclear today. HSPB2 is mainly expressed in skeletal and heart muscles [23]. It is also called MKBP (myotonic dystrophy protein kinase binding protein) as it binds to myotonic dystrophy protein kinase (DMPK), increasing its activity and conferring thermal protection [24]. HSPB2 interacts mainly with HSPB3 (also referred to as HSP17) [25], and together these molecules participate in muscle differentiation [26,27,28]. Mutations of both HSPB2 and HSPB3 have been linked with several neurodegenerative and neuromuscular diseases [29,30]. Several studies by us and others have revealed that sHSPs play an important role in oncogenesis and malignant progression and are potential biomarkers for cancer diagnosis, prognosis, and therapeutic targets [16,30,31,32,33,34]. Given that the involvement of *HSPB2* and *HSPB3* genes in bladder tumorigenesis remains unclear, we herein analyze *HSPB2* and *HSPB3* gene expression levels in bladder tumors and matched adjacent normal urothelium. Further, we evaluated their association with post-treatment disease course, in order to assess their potential role in disease prognosis and management.

## 2. Results

### 2.1. Upregulation of HSPB2 and HSPB3 Gene Expression Levels in Human BlCa Cell Lines

Firstly, the expression levels of *HSPB2* and *HSPB3* genes were examined in BlCa cell lines of gradually increasing malignancy in order to evaluate their expression according to tumor grade. Both *HSPB2* and *HSPB3* gene expression levels were upregulated in the advanced tumor grade cell line TCCSUP (Figure 1A,B). This finding was also largely confirmed for HSPB2 by immunoblotting analysis (Figure 1C).

### 2.2. Baseline Clinical Data

Given the aforementioned findings in BlCa cell lines which indicate that following an initial downregulation, *HSPB2* and *HSPB3* genes are induced in advanced tumors, their expression levels were investigated in bladder urothelium biopsies. Most of the patients were males (81%) and the median patients’ age was 67.5 years old. Focusing on disease features, 62 and 38 of the patients suffered from NMIBC (TaT1) and MIBC (T2–T4), respectively, while 60% of the tumors were characterized as high grade (HG) according to the WHO 2004 guidelines [35,36]. Moreover, 9.7%, 32.3% and 58.1% of NMIBC patients were stratified as low, intermediate, and high risk, respectively, according to EORTC guidelines [37]; detailed patients’ clinicopathological characteristics are presented in Table 1.

Follow-up was completed for 89% of the patients, whereas 11 patients, six NMIBC (9.7%) and five MIBC (13.2%) patients were excluded due to insufficient monitoring data. The median follow-up time (reverse Kaplan–Meier method) is 27.0 months (95% CI: 23.12–30.88). Regarding NMIBC (TaT1), disease relapse was reported in 21 (37.5%) of the 56 followed-up patients and the mean disease-free survival (DFS) was 49.84 months (95% CI: 41.29–58.38), while overall survival (OS) of MIBC (T2–T4) was 34.16 months (95% CI: 28.27–40.05).

### 2.3. Reduced Expression Levels of the HSPB2 and HSPB3 mRNA in Bladder Tumors Compared with Paired Non-Cancerous Tissues

The comparison of *HSPB2* and *HSPB3* mRNA levels between pooled bladder tumors and their matched normal adjacent counterparts revealed a prominent downregulation of the expression of these molecular chaperones in the vast majority of the tumors (Figure 2) indicating a generalized suppression during cancer initiation and progression. Moreover, ROC curve analysis confirmed the ability of *HSPB2* and *HSPB3* expression to discriminate bladder tumors from normal urothelium, also highlighting the superior value of *HSPB2* (AUC: 0.815, 95% CI: 0.740–0.891, *p* < 0.001) as compared to *HSPB3* gene expression levels (AUC: 0.659, 95% CI: 0.568–0.750, *p* = 0.003).

### 2.4. HSPB2 and HSPB3 mRNA Expression Levels Are Related to Unfavorable Prognostic Features of BlCa

The noted downregulation of *HSPB2* and *HSPB3* mRNA expression levels in bladder tumors urged us to examine their potential association with the clinicopathological features of BlCa. Thus, the expression levels of these *s-HSPs* genes were investigated according to tumor stage and grade. Higher expression levels of *HSPB2* (*p* = 0.018) and *HSPB3* (*p* < 0.001) were found in MIBC vs. the NMIBC tumors (Figure 3A,B). Moreover, consistent with our findings in BlCa cell lines (Figure 1), increased *HSPB2* and *HSPB3* gene expression levels were found in tumors of a higher stage (Figure 3C,D) (*HSPB2*, *p* = 0.009; *HSPB3*, *p* = 0.002) and grade (Figure 3E,F) (*HSPB2*, *p* = 0.054; *HSPB3*, *p* = 0.011).

### 2.5. Low Expression Levels of HSPB2 and HSPB3 mRNA Are Correlated with Higher Risk for NMIBC (TaT1) Patients’ Short-Term Relapse

Since the expression of *HSPB2* and *HSPB3,* mRNA was associated with prognostic features of BlCa, their significance for patients’ treatment outcome was analyzed. The survival analysis of the NMIBC cohort was performed using disease recurrence as a clinical endpoint event for DFS. Kaplan–Meier survival curves highlighted the significantly shorter DFS of TaT1 patients with lower expression of either *HSPB2* (*p* = 0.039) or *HSPB3* (*p* = 0.033) gene (Figure 4A,B), indicating that during the early phases of the disease high expression levels of these chaperones are likely favorable for the patient. The univariate Cox regression analysis confirmed the higher risk for disease relapse of TaT1 patients under expressing *HSPB2* (HR: 2.527; 95% CI: 1.004–6.361) or *HSPB3* (HR: 3.489; 95% CI: 1.018–11.96) (Table 2). Consistently, the higher risk for short-term relapse and poorer DFS was highlighted for TaT1 patients with a decreased expression of both *HSPB2* and *HSPB3* mRNA levels, which was confirmed by Kaplan–Meier curves (*p* = 0.033) (Figure 4C) and the univariate Cox analysis (HR: 2.132; 95% CI: 1.143–3.984; *p* = 0.017). The multivariate Cox regression analysis adjusted for the major clinical prognostic factors of BlCa highlighted the independent clinical value of *HSPB2* and *HSPB3* mRNA expression for the outcome of NMIBC. More precisely, multivariate Cox models confirmed the unfavorable DFS of TaT1 patients under expressing *HSPB2* (HR: 3.101; 95% CI: 1.134–8.484; *p* = 0.027) or *HSPB3* (HR: 4.872; 95% CI: 1.322–17.96; *p* = 0.017), independently to tumor stage, grade, patient’s age and EORTC risk stratification (Table 2).

Most interestingly however, a trend [not statistically significant, *p* > 0.05; Cox regression analysis for MIBC (T2–T4) patients’ overall survival (OS) following treatment (Appendix A)] for an inverted association of *HSPB2* and (to a far lesser extend) *HSPB3* genes expression levels with patients’ OS was noticed in advanced stages of the diseases (MIBC; T2–T4). Specifically, it was observed that those patients expressing high levels of these s*HSP* genes showed a trend for a higher risk regarding a more rapid evolvement of the disease and thus reduced OS (Figure 5).

### 2.6. HSPB2 and HSPB3 Knockdown Decreases Cell Viability and Enhances Chemosensitivity in Human Bladder Cancer Cells

Given our findings showing the likely implication of HSPB2 and HSPB3 in BlCa, then the effect of *HSPB2* and *HSPB3* genes knockdown in BlCa cell lines was evaluated by applying RNAi-mediated gene expression silencing (Figure 6A,B). To analyze the potential effect of *HSPB2* and *HSPB3* genes knockdown on BlCa cells survival, cell viability assay after siRNAs transfections was performed in the HTB9, T24 and TCCSUP BlCa cell lines. Results showed that *HSPB2* knockdown significantly reduced cell viability (*p* < 0.01) in T24 and TCCSUP cells; notably, *HSPB3* knockdown had no considerable effect on BlCa cells survival (Figure 6C).

Further was investigated whether knockdown of the *HSPB2* and *HSPB3* genes exerts a synergistic effect with chemotherapy. To this end, cells pre-treated with RNAi oligonucleotides were subjected to 200 nM of DXR. *HSPB2* and *HSPB3* knockdown were found to increase remarkably the sensitivity of the three tested BlCa cell lines to DXR (*p* < 0.01) (Figure 6C). To further verify these findings, apoptosis rate and cell death was examined in the TCCSUP BlCa cell line of tumor grade III. Results showed an enhancement, vs. cells being exposed to only RNAi or DXR, of Annexin^+^/PI^+^ stained cells after RNAi/DXR treatment (Figure 6D), further supporting the pro-survival role of sHSPs during DXR treatment.

## 3. Discussion

Urothelial bladder carcinoma is one of the most frequent types of cancer worldwide and is considered a disease with a great heterogeneity, not only at molecular level, due to the accumulation of a plethora of genetic mutations and/or epigenetic changes, but also regarding patients’ prognosis and treatment outcome [38]. Most bladder tumors are superficial during diagnosis and thus a main issue is the potential risk for recurrence and progression into a muscle-invasive disease [39]. Despite the progress in diagnosis, medical achievements, chemotherapy and irradiation, MIBC is still associated with a poor prognosis [40]. Due to the special features of this disease, the combinational use of different molecular targets could constitute an alternative approach for the accurate diagnosis and prognosis of patients.

Recently, growing evidence by us and others has shown that sHSPs expression is frequently deregulated in diverse malignancies including colorectal, breast, lung and pancreatic cancer [33,34,41,42]. It is well established that sHSPs play a pivotal role in the development and progression of cancer and, more specifically, they have been associated with several hallmark features of cancer, including tumorigenesis, cell growth, apoptosis, metastasis and chemoresistance [13,17,28]. Therefore, sHSPs have been proposed as potential clinical biomarkers for diagnosis and prediction of prognostic outcomes in cancer patients [43,44].

Given the fact that bladder is a hollow smooth muscle organ lined by a mucous membrane [45], and while it is known that HSPB2 is mainly expressed in skeletal and smooth muscles, where it forms oligomers with HSPB3 [46], to the best of our knowledge the potential role of HSPB2 and HSPB3 in tumorigenesis or progression of BlCa has not been studied yet. Thus, *HSPB2* and *HSPB3* expression in bladder tumors and cell lines was assessed and their potential clinical significance concerning patients’ survival and disease outcome. To this end, four BlCa cell lines were used; each one reflecting a distinct malignancy grade (I, II, III, and IV) and their normal control (human primary bladder epithelial cells; BlEc). *HSPB2* and *HSPB3* gene expression levels were upregulated in advanced tumor grade cell lines. This result was also confirmed by protein analysis, where HSPB2 was upregulated in the advanced tumor grade cell line TCCSUP. Additionally, from our analyses, a relative decrease in the *HSPB2* and *HSPB3* mRNA expression has been noticed in cancerous as compared to adjacent normal tissues, whereas increased *HSPB2* and *HSPB3* mRNA levels were observed in MIBC (T2–T4) compared to superficial (TaT1) tumors and in high vs. low grade tumors. This outcome supports the critical role of these chaperones in disease progression and their discrete regulation in superficial and muscle invasive tumors. In line with these findings, decreased levels of sHSPs have been identified in colorectal, pancreatic and renal malignancies [26], while increased levels of *HSPB2* expression were found in breast cancer where it was proposed that HSPB2 possesses an anti-apoptotic role via inhibiting the extrinsic apoptotic pathway [47]. Moreover, it was recently reported that *HSPB3* mRNA expression is upregulated in lung squamous cell carcinoma tissues compared with normal lung tissues based on the TCGA database. Consequently, the role of HSPB2 and HSPB3 in tumor progression could be associated with suppression of malignant cells spontaneous apoptosis.

Considering the differences in the clinical outcome of superficial from muscle invasive BlCa, a survival analysis was conducted individually. Focusing on superficial tumors (TaT1), decreased *HSPB2* and *HSPB3* mRNA levels were correlated with a significantly higher risk for disease recurrence following TURBT, independently of tumor stage, grade, EORTC-risk group, and gender. These data correlate nicely with data in the Human Protein Atlas database, where high expression levels of *HSPB2* mRNA were found to correlate with lower risk of short-term relapse in patients. This finding can be likely explained by the fact that, as mentioned before, urothelial bladder carcinoma is a highly heterogeneous disease [3,48] since it contains two subgroups with very distinct molecular features and different potential pathways of pathogenesis [48]. As shown by Zhang et al., there are different effects of HSPs in different subtypes of each cancer, since for example in breast cancer HSPB1 is positively associated with inflammation in basal type but negatively associated with inflammation in luminal B, so it is very important to define the different subtypes of cancer, especially in highly heterogenous types of bladder cancer [49].

It is well established that high-risk NMIBCs patients have a high possibility for disease recurrence and/or progression and treatment failure [5,6]. In 2020, the EAU highlighted the need for reliable biomarkers for high-risk NMIBC patients to receive the correct treatment without delay in order to improve the predictive accuracy [50].

The inverse association between *HSPB2* and *HSPB3* expression levels and disease outcomes in TaT1 patients is in accordance with previous findings for HSP70 and HSP90 [40,51]. This notable finding highlights the biphasic action of sHSP genes during carcinogenesis (see Graphical Abstract). Specifically, whereas high sHSPs levels can delay tumor progression at early phases of tumorigenesis by ensuring proteome stability and are thus protective, at advanced stages high levels of sHSPs offer a significant advantage to tumor cells by suppressing tumor cells increased proteotoxic stress (caused, among others, by extensive genomic instability) correlating thus with increased tumor aggressiveness [13]. In support of this hypothesis, it has been shown [52] that another HSP named Clusterin can act simultaneously as a tumor suppressor and a tumor promoter depending on the stage of cancer that is overexpressed. The proposed dual function of these chaperones in tumor evolvement and their late induction during tumorigenesis that results in the so-called non-oncogenic addiction is currently under investigation [53].

Increasing evidence supports that HSPs play a critical role in the regulation of immune responses [54]. Notably, Bendz et al. have demonstrated that HSP70 enhances tumor antigen presentation through complex formation and intracellular antigen delivery without innate immune signaling and finally activation of cytotoxic CD8^+^ T cells [55]. Furthermore, lung cancer patients with tumors expressing high *HSPB3* mRNA levels had a better prognosis, while the *HSPB3* gene was strongly correlated with CD8^+^ T cell infiltration [56]. Therefore, declined HSPs functionality in non-metastatic BlCa tumors could further decrease the immune responses against cancer cells. The reduced BCG antigens load presented by antigen presenting cells could thus result in therapy resistance and a poor clinical outcome. More interestingly, this correlation tended to be inverted in advanced stages of the disease (muscle-invasive tumors), in line with a suggested biphasic role of sHSPs in advanced tumorigenesis where, by suppressing increased proteotoxic stress, they confer a significant advantage in tumor cells that overexpress these chaperones [13,57]. In support, as reported before, *HSPB2* and *HSPB3* were expressed in high levels in high grade BlCa cell lines where they suppress DXR-toxicity since their downregulation was found to increase its toxicity in BlCa cell lines. Moreover, flow cytometry confirmed an enhance in apoptosis rate in advanced tumor grade cell line TCCSUP, when chemotherapy agent DXR, was combined with the silencing of *HSPB2* and *HSPB3.* Supportively, HSPB2 was found in previous studies to possess an anti-apoptotic role [49], and it is well established that deficiency in apoptotic pathways is a hallmark of many cancer types [58]; thus, therapeutic strategies that would target the apoptotic pathways could possibly constitute an effective therapeutic approach [59]. Consistently, HSP inhibitors have shown promising results in cancer treatment [60]. Nevertheless, our study is characterized by some limitations that need to be addressed. Firstly, our cohort size is of medium size and the patients’ cohort was not equivalently stratified in the defined subgroups, which could diminish the obtained findings. Future studies should be conducted to further evaluate the role of HSPB2 and HSPB3 in BlCa prognosis.

Overall, given the extensive clinical heterogeneity of bladder malignancies and the fact that NMIBC and MIBC tumors display different mechanisms of pathogenesis [51], our novel finding of the biphasic impact of *HSPB2* and to a lesser extent *HSPB3* genes expression levels in early vs. advanced carcinogenesis, adds further knowledge towards the more accurate stratification of the different BlCa stages.

## 4. Materials and Methods

### 4.1. Cell Lines and Cell Culture Conditions

The human normal primary bladder epithelial cells (BlEc), transitional cell carcinoma lines RT4 (grade I), HTB9 (grade II), T24 (grade III) and TCCSUP (grade IV) were purchased from the American Tissue Culture Collection (ATCC, Manassas, VA, USA). Cells were cultured in bladder epithelial cell basal medium (PCS-420-032, ATCC) (BlEc) or Dulbecco’s modified Eagle’s medium (11965092, Thermo Fisher Scientific Inc., Waltham, MA, USA), supplemented with 10% (*v*/*v*) fetal bovine serum, and 2 mM L-Glutamine (RT4, HTB9, T24, TCCSUP) in a humidified incubator of 5% CO_2_ at 37 °C. Doxorubicin (DXR) was obtained from Merck KGaA (Darmstadt, Germany).

### 4.2. siRNA Transfection

For RNAi analyses, cells were transfected by using the SMARTpool ON-TARGET plus HSPB2 siRNA (L-008852-00-0005), the SMARTpool ON-TARGET plus HSPB3 siRNA (L-008830-00-0005) or the ON-TARGETplus non-targeting pool (Scramble) (D-001810-10-05) (GE Healthcare Dharmacon Inc., Lafavette, CO, USA) All transfections were carried out using Lipofectamine 2000 (Invitrogen, Paisley, UK), according to the manufacturer’s instructions.

### 4.3. Cell Survival Assay

Cells were seeded onto 96-well plates (Corning, Inc., New York, NY, USA) at a density of 5000 cells/well and were transfected with siRNAs. Transfected cells were treated with DXR (Pfizer Hellas S.A) at the concentration of 200 nM after 48 h of transfection. Following incubation for 24 h, the medium was replaced by 3-(4,5-dimethylthiazol-2-yl)-2,5-diphenyltetrazolium bromide (M5655, Sigma-Aldrich, Taufkirchen, Germany) dissolved at a final concentration of 1 mg/mL in serum-free, phenol red-free medium. The formed formazan crystals were then dissolved by isopropanol and the absorbance of the solution was measured at a 570 nm wavelength.

### 4.4. Apoptosis Assay

To determine apoptosis induced in cells after transfection with HSPB2 or HSPB3 RNAi oligonucleotides and DXR treatment, the FITC Annexin V Apoptosis Detection Kit I (556547, BD Pharmingen™, Heidelberg, Germany) was used, according to the manufacturer’s recommendations. After staining with FITC-Annexin V and Propidium Iodide (PI), samples were analyzed by a BD Accuri™ C6 Plus Flow Cytometer (BD Biosciences, Heidelberg, Germany,) equipped with BD Accuri™ C6 Plus software (BD Biosciences).

### 4.5. Patients Cohort

In the present study, fresh-frozen bladder tumors were obtained from 100 BlCa patients treated with TURBT for primary NMIBC or RC for primary MIBC at the ‘Laiko’ General Hospital, Athens, Greece. Moreover, from 49 patients of the cohort, adjacent-normal bladder specimens were obtained following pathologist’s evaluation for the absence of dysplasia or carcinoma in situ (CIS). Bladder specimens were incubated in RNAlater Solution (AM7020, Ambion, Inc., Austin, TX, USA), according to manufacturer’s instructions, and stored at −80 °C. None of the patients received neoadjuvant treatment prior to surgery, while adjuvant therapy was applied according to the European Association of Urology (EAU) guidelines.

NMIBC patients’ (TaT1) follow-up included cystoscopy and urinary cytology (for high-grade tumors) according to EAU guidelines. MIBC patients (T2–T4) were followed-up by renal ultrasound (at 3 months) and thoracoabdominal computed tomography (CT) magnetic resonance imaging (MRI) (every 6 months), while additional kidney ultrasound and thoracoabdominal CT/MRI, bone scan or brain MRI were performed following symptoms. Disease recurrence was confirmed by TURBT, following a positive follow-up cystoscopy, or CT for NMIBC and MIBC patients, respectively.

The study was approved by the Ethics Committee of “Laiko” General Hospital, Athens, Greece and conducted according to the 1975 Declaration of Helsinki ethical standards, as revised in 2008. Informed consent was obtained by all the patients who participated.

### 4.6. Total RNA Extraction

Bladder tissue specimens (40–100 mg) were cryo-pulverized mechanistically by a tissue pulverizer using liquid nitrogen. Total RNA was isolated from pulverized specimens using TRI-Reagent (TR 118, Molecular Research Center, Cincinnati, OH, USA). Total RNA from BlCa cell lines was extracted using RNAiso plus (Takara Bio Inc., Shiga, Japan). RNA concentration and purity were evaluated spectrophotometrically at 260 and 280 nm. Agarose gel electrophoresis was used for the assessment of RNA quality.

### 4.7. First-Strand cDNA Synthesis

First-strand cDNA synthesis was performed in a 20 μL reaction containing 1 μg total RNA, 0.5 mM dNTPs mix, 50 U M-MLV Reverse Transcriptase (28025-013, Invitrogen, Carlsbad, CA, USA), 40 U RNaseOUT Recombinant Ribonuclease Inhibitor (10777-019, Invitrogen), and 5 μM oligo(dT) primer, at 37 °C for 60 min. Thereafter, MMLV was inactivated at 70 °C for 15 min.

### 4.8. Quantitative Real-Time Polymerase Chain Reaction

*HSPB2* and *HSPB3* mRNA expression levels were quantified by an EvaGreen-based qPCR assay. Gene-specific primers were designed for *HSPB2*, *HSPB3*, as well as for housekeeping gene encoding hypoxanthine phosphoribosyltransferase 1 (*HPRT1*), according to their NCBI published mRNA sequences (NCBI RefSeq accession number: NM_001541.4, NM_006308.3 and NM_000194.3, respectively). Specific pairs of primers were designed according to the cDNA sequences of *HSPB2*, *HSPB3* and *HPRT1*. The sequences of *HSPB2* primers were 5′-CCGAGTACGAATTTGCCAACC-3′ and 5′-AGGCCGGACATAGTAGCCAT-3′, of *HSPB3* 5′-CCAGTGCGTTACCAGGAAGA-3′ and 5′-TGGTTTTCCTCAGGTCCACG-3′ and those of *HPRT1* primers were 5′-TTGGAAAGGGTGTTTATTCCTCAT-3′ and 5′-ATGTAATCCAGCAGGTCAGCAA-3′.

To carry out qPCR analyses, the 5× HOT FIREPol^®^ EvaGreen^®^ qPCR Supermix (08-25-00001, Solis BioDyne, Tartu, Estonia) and PikoReal^TM^ Real-Time PCR System (Thermo Fisher Scientific Inc., Waltham, MA, USA) were used. *HSPB2* and *HSPB3* expression was quantified by the comparative CT (2^−ΔΔCT^) relative quantification method [61], using *HPRT1* as an endogenous reference control for normalization purposes, and the BJ cells (normal foreskin human fibroblasts) as our assay calibrator. In this regard, as 1 arbitrary unit (au) was defined, the target gene expression of BJ cells following normalization (ΔCT^BJ^ = CT^test gene^ − CT^HPRT1^) and thereafter the normalized expression of the tested samples (ΔCT^sample^ = CT^test gene^ − CT^HPRT1^) was quantified relative to the calibrator by the 2^−ΔΔCT^ method, where ΔΔCT = ΔCT^sample^. The cycling steps were the following: (1) 3 min at 95 °C for denaturation; (2) 40 cycles consisting of two steps—3 s at 95 °C in order for the PCR products to denaturate and 30 s at 60 °C for primer annealing and extension; (3) generation of a melting curve, aiming to distinguish the main product from the primer dimers and other non-specific products; the latter usually have a lower T_m_ (<75 °C) than the main product.

### 4.9. Immunoblotting Analysis

Cells were lysed on ice in NP-40 lysis buffer (150 mM NaCl, 1% NP-40, 50 mM Tris pH 8.0) containing protease and phosphatase inhibitors and lysates were cleared by centrifugation for 10 min at 19,000× *g* (4 °C). The total protein content of each lysate was measured by Bradford assay (Bio-Rad). Equal total protein μg per sample were separated by SDS-PAGE and blotted onto a nitrocellulose membrane (Immobilion-P, Millipore, Eschborn, Germany). Primary and horseradish peroxidase-conjugated secondary antibodies were applied for 1 h at room temperature and immunoblots were developed by an enhanced chemiluminescence reagent kit (sc-2048, Santa Cruz Biotechnology). Primary antibodies against HSPB2 (sc-514154) and GAPDH (sc-25778), diluted 1:1000 in blocking buffer were purchased from Santa Cruz Biotechnology, Inc. (Dallas, TX, USA). Secondary antibodies were diluted 1:3000 in blocking buffer and purchased from Jackson ImmunoResearch Europe Ltd. (Ely, UK). Immunoblots quantitation was performed by scanning densitometry and ImageJ software (National Institutes of Health, USA).

### 4.10. Statistical Analysis

IBM SPSS Statistics 20 (IBM Corp., Armonk, New York, NY, USA) was used for the statistical analysis. To evaluate the normal distribution of the data, Shapiro–Wilk and Kolmogorov–Smirnov tests were applied. When the sample distribution was normal, the parametric unpaired *t*-test with a two-tailed *p*-value was applied. In non-normal distribution, the non-parametric Mann–Whitney U and Kruskal–Wallis tests were used appropriately to study the association of *HSPB2* and *HSPB3* gene expression levels with the clinicopathological data of BlCa patients. ROC curve analysis was performed to test the value of *HSPB2* and *HSPB3* expression in discriminating bladder tumors from normal bladder specimens.

Kaplan–Meier survival curves, using log-rank test, and Cox logistic regression analysis were applied for the survival analysis of the patients. In this regard, post-treatment disease relapse and patients’ death were used as clinical end-point events for the NMIBC (TaT1) and MIBC (T2–T4) patients, respectively, while X-tile algorithm was used for the adoption of an optimal cut-off value of *HSPB2* and *HSPB3* gene expression levels. Finally, internal validation of Cox regression models was performed by bootstrap analysis based on 1000 bootstrap samples.

## 5. Conclusions

The expression levels of *HSPB2* and *HSPB3* molecular chaperone genes in BlCa patients are significantly deregulated, being thus able to distinguish malignant (at different stages) specimens from adjacent normal bladder tissues. Muscle-invasive and high-grade tumors exhibit increased levels of these cytoprotective chaperones in comparison with non-muscle invasive and low-grade tumors. Moreover, regarding the treatment outcome, the downregulation of *HSPB2* and *HSPB3* genes in early tumors is significantly associated with BlCa recurrence in NMIBC patients, whereas increased expression of *HSPB2* (and likely *HSPB3*) genes likely correlate with decreased overall survival in advanced disease stages (muscle-invasive tumors). These genes can be combined with other biomarkers to improve the estimation of patient’s status.They can also constitute novel therapeutic targets in the treatment of BlCa.

## Figures and Tables

**Figure 1 ijms-24-02609-f001:**
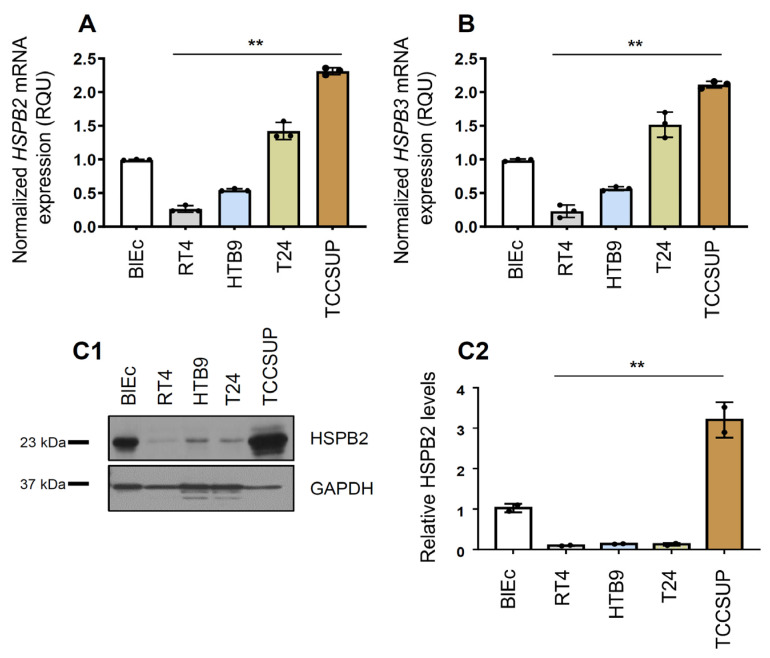
(**A**,**B**) Relative *HSPB2* and *HSPB3* mRNA expression levels in BlCa cell lines of increasing malignancy vs. BlEc cells; genes’ expression in BlEc cells was set to 1. (**C1**) Representative immunoblotting analysis of HSPB2 in BlCa cell lines vs. B1Ec cells. Probing with GAPDH was used as total protein loading reference. (**C2**) Relative protein quantification of HSPB2 in BlCa cell lines vs. B1Ec cells; proteins’ expression in BlEc cells was set to 1. *p*-value was calculated using unpaired *t*-test. Bars, ±SD; ** *p* < 0.01.

**Figure 2 ijms-24-02609-f002:**
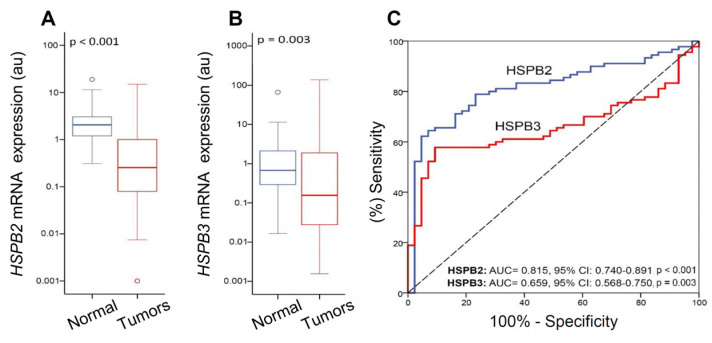
(**A**,**B**) Comparison of *HSPB2* (**A**) and *HSPB3* (**B**) mRNA expression levels in BlCa specimens vs. normal urothelium. *p*-value was calculated using the Mann–Whitney *U* test. (**C**) ROC curve analysis of *HSPB2* and *HSPB3* genes expression levels’ ability to discriminate bladder tumors from normal urothelium. The middle lines inside the boxes indicate the median (50th percentile or second quartile), while the lower and the upper box boundaries represent the 25th percentile (first quartile) and the 75th percentile (third quartile), respectively. The lower and upper whisker extend to the lowest and highest value, respectively, within the 1.5× interquartile range (box height) from the box boundaries. *p*-value was calculated by the Hanley and McNeil method.

**Figure 3 ijms-24-02609-f003:**
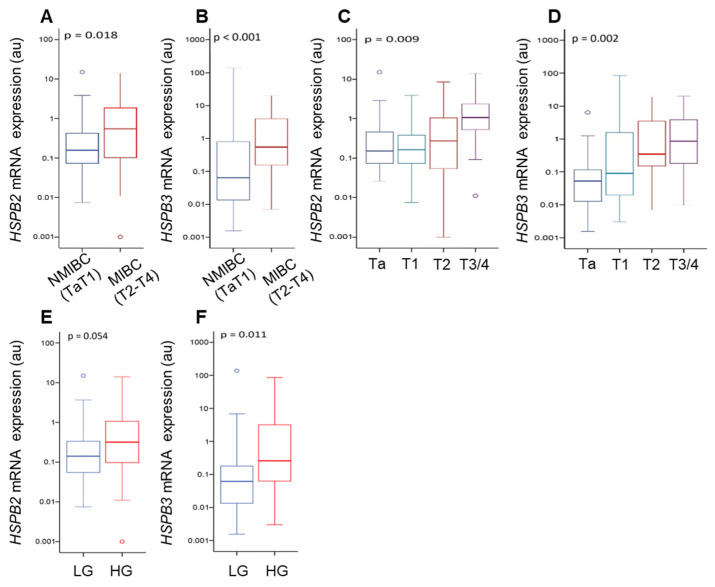
*HSPB2* and *HSPB3* mRNA expression analysis in BlCa patients. Box plots presenting the correlation of *HSPB2* and *HSPB3* genes expression in the screening cohort (*n* = 100) with non-muscle invasive (NMIBC) and muscle invasive (MIBC) bladder cancer (**A**,**B**), as well as with tumor stage (**C**,**D**) and tumor grade (**E**,**F**). The middle lines inside the boxes indicate the median (50th percentile or second quartile), while the lower and the upper box boundaries represent the 25th percentile (first quartile) and the 75th percentile (third quartile), respectively. The lower and upper whisker extend to the lowest and highest value, respectively, within the 1.5× interquartile range (box height) from the box boundaries. *p*-value was calculated using the Mann–Whitney *U* (**A**,**B**,**E**,**F**) and Kruskall–Wallis (**C**,**D**) tests.

**Figure 4 ijms-24-02609-f004:**
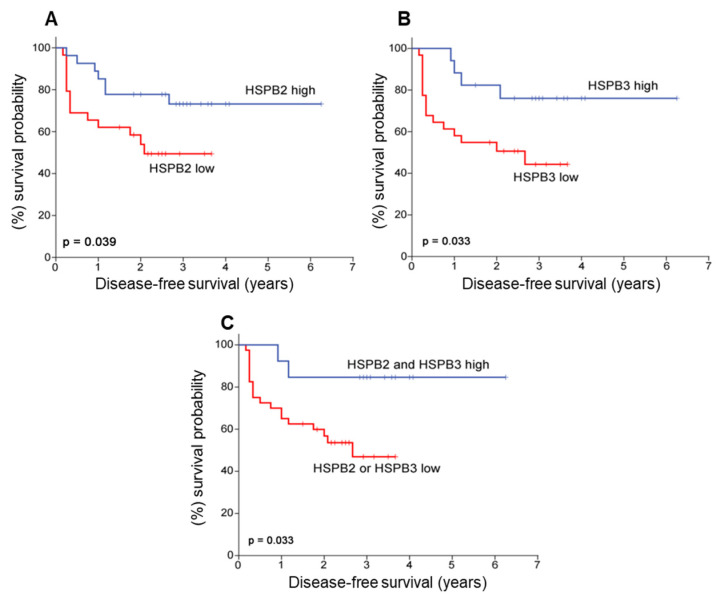
Kaplan–Meier survival curves for the disease-free survival (DFS) of NMIBC (TaT1) patients according to *HSPB2* (**A**), *HSPB3* (**B**) and both (**C**) mRNA expression levels. *p*-value was calculated using the log-rank test.

**Figure 5 ijms-24-02609-f005:**
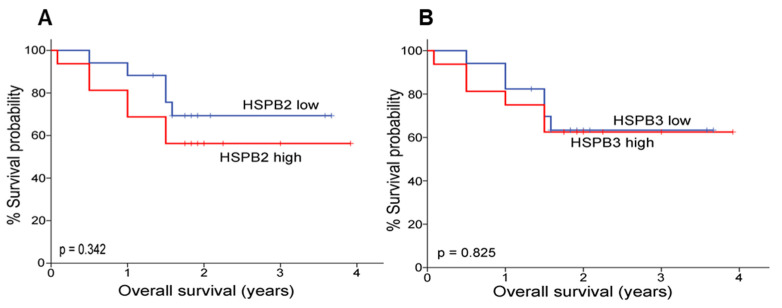
Kaplan–Meier survival curves for the overall survival (OS) of MIBC (T2–T4) patients according to *HSPB2* (**A**) and *HSPB3* (**B**) mRNA expression levels. *p*-value was calculated using the log-rank test.

**Figure 6 ijms-24-02609-f006:**
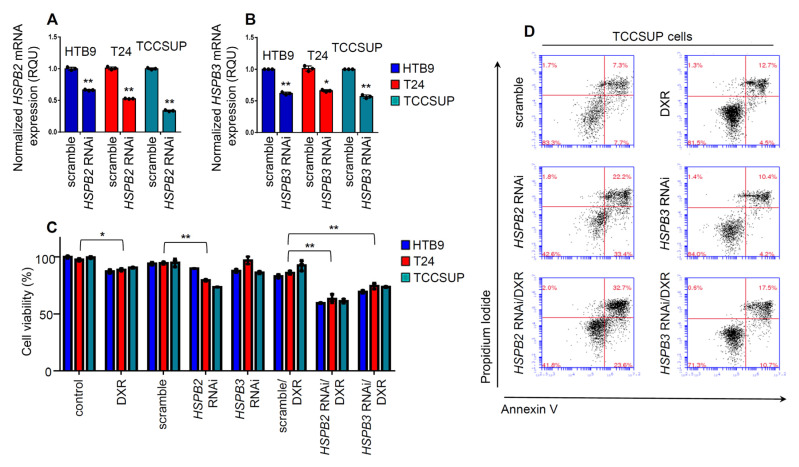
(**A**,**B**) Relative *HSPB2* (**A**) and *HSPB3* (**B**) mRNA expression levels in the indicated BlCa cell lines after transfection with the shown RNAi oligonucleotides or a non-targeting pool (scramble) for 48 h. (**C**) Cell viability (MTT assay) of BlCa cell lines being treated with RNAi oligonucleotides or a non-targeting pool (scramble) for 48 h and then exposed to DXR (200 nM) for 24 h. (**D**) Flow Cytometry after transfecting the TCCSUP cell line with *HSPB2* or *HSPB3* RNAi oligonucleotides or a non-targeting pool (scramble) for 48 h and then exposed to DXR (200 nM) for 24 h. The percentages of Annexin V- and PI- positive cells are indicated. In (**A**,**B**) control samples values were set to 1; in (**C**) control samples values were set to 100. *p*-value was calculated using an unpaired *t*-test. Bars, ±SD; * *p* < 0.05; ** *p* < 0.01.

**Table 1 ijms-24-02609-t001:** Clinicopathological features of the BlCa patients.

Variable	No. of Patients *n* = 100
**Disease**	
NMIBC (Ta, T1)	62 (62%)
MIBC (T2–T4)	38 (38%)
**Tumor stage**	
pTa	30 (30%)
pT1	32 (32%)
pT2	21 (21%)
pT3	10 (10%)
pT4	7 (7%)
**Grade (WHO 2004)**	
Low	40 (40%)
High	60 (60%)
**Grade (WHO 1973)**	
1	7 (7%)
2	37 (37%)
3	56 (56%)
**Gender**	
Male	81 (81%)
Female	19 (19%)
**Non-muscle invasive bladder cancer (NMIBC; TaT1)**	
**EORTC risk group**	
Low risk	6 (9.7%)
Intermediate risk	20 (32.3%)
High risk	36 (58.1%)
**Disease monitoring**	
Follow-up patients	56
Disease-free	35 (62.5%)
Recurrence	21 (37.5%)
Excluded from follow-up	6
**Muscle-invasive bladder cancer (MIBC; T2–T4)**	
**Disease monitoring**	
Follow-up patients	33
Alive	21 (63.6%)
Death	12 (36.4%)
Excluded from follow-up	5

**Table 2 ijms-24-02609-t002:** Cox regression analysis for the prediction of NMIBC (TaT1) patients’ risk for relapse (DFS) following treatment.

Covariant	NMIBC (TaT1) Disease-Free Survival (DFS)				
	Univariate Analysis				
	HR ^a^	95% CI ^b^	*p*-Value ^c^	Bootstrap	Bootstrap
				BCa 95% CI ^d^	*p*-Value ^c^
** *HSPB2* **					
High expression	1				
Low expression	2.527	1.004–6.361	0.049	0.970–8.795	0.03
** *HSPB3* **					
High expression	1				
Low expression	3.489	1.018–11.96	0.047	1.022–46.79	0.02
**Tumor Stage**					
Ta	1				
T1	1.271	0.708–2.280	0.422	0.727–2.120	0.44
**Tumor Grade**					
Low	1				
High	1.281	0.708–2.316	0.413	0.681–2.421	0.38
**Age** (Continuous variable)	1.002	0.972–1.033	0.896	0.973–1.031	0.88
**EORTC risk group** Low vs. Interm. vs. High	1.138	0.740–1.750	0.556	0.749–1.826	0.55
	**Multivariate analysis for *HSPB*** ***2* ^e^**				
**Covariant**	**HR^a^**	**95% CI ^b^**	** *p* ** **-value ^c^**	**Bootstrap**	**Bootstrap**
				**BCa 95% CI ^d^**	** *p* ** **-value ^c^**
** *HSPB2* **					
High expression	1				
Low expression	3.101	1.134–8.484	0.027	0.732–36.59	0.03
**Tumor Stage**					
Ta	1				
T1	2.67	0.497–14.36	0.252	0.346–105.7	0.28
**Tumor Grade**					
Low	1				
High	0.976	0.270–3.520	0.97	0.160–5.416	0.98
**Age** (Continuous variable)	1.001	0.950–1.054	0.966	0.941–1.084	0.97
**EORTC risk group** Low vs. Interm. vs. High	0.591	0.179–1.949	0.388	0.136–2.179	0.35
	**Multivariate analysis for *HSPB3* ^e^**				
** *HSPB3* **					
High expression	1				
Low expression	4.872	1.322–17.96	0.017	1.00–1.22 × 10^6^	0.02
**Tumor Stage**					
Ta	1				
T1	3.404	0.620–18.67	0.158	0.053–1.18 × 10^7^	0.19
**Tumor Grade**					
Low	1				
High	0.654	0.162–2.635	0.55	0.084–1.876	0.63
**Age** (Continuous variable)	0.977	0.933–1.023	0.32	0.931–1.020	0.35
**EORTC risk group** Low vs. Interm. vs. High	0.781	0.239–2.553	0.682	0.063–2.45 × 10^4^	0.63

^a^ Hazard Ratio. ^b^ 95% confidence interval of the estimated HR. ^c^ Calculated by test for trend. Bootstrap *p*-value is based on 1000 bootstrap samples. ^d^ Bootstrap bias-corrected and accelerated 95% confidence interval of the estimated HR based on 1000 bootstrap samples. ^e^ Multivariate analysis adjusted for tumor stage, tumor grade, EORTC-risk group stratification, patients’ age.

## Data Availability

The data presented in this study are available on request from the corresponding author.

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
