# Peer review of "Evaluation of the Small Heat Shock Protein Family Members HSPB2 and HSPB3 in Bladder Cancer Prognosis and Progression"

_ijms, 2023, doi:10.3390/ijms24032609_

Round 1

Reviewer 1 Report

In this manuscript, several concerns are to be addressed as follows:

1.      The abstract lacks quantitative information about the research methods, and the presentation is confusing. There is a paucity of detail given about the experimental protocols and which metrics were calculated. The authors could shorten the background and give more details on the methods used.

2.      Results:

-          All introductory paragraphs describing methods used like lines 86-89, and 101-105 should be either transferred to the discussion or deleted and directly present the results.

-          Also, all paragraphs with references should be transferred to the discussion section (E.g. lines 124-126).

3.      The discussion needs to be deepened by more interpretations of the study findings and correlate the relation of the estimated parameters. Paragraphs (lines 251-278) should exist in the introduction, not in the discussion section. Also, the authors should clarify the limitation of the study.

4.      Lines 394-400: separate into short sentences. Clarify the type of sample at the beginning of the paragraph.

5.      The writing style should be formal from the third-person perspective. Do not use we or our (E.g. line 21, our initial, line 25, we observed).

6.      It is not preferable to begin sentences with abbreviations like BICa in line 51.

Reviewer 2 Report

The manuscript studied the role of HSPB2 and HSPB3 molecular chaperone genes in bladder cancer. The study used both in vitro BlCa cell lines and in vivo bladder urothelium biopsies. The authors concluded dysregulation of  HSPB2 and HSPB3 molecular chaperone genes in BlCa. Additionally, their levels were linked to aggressiveness of cancer, survival, and treatment efficacy, suggesting potential diagnostic and prognostic roles. 

My comments

-The authors evaluated only HSB2 by western blot in vitro, why was not it evaluated in vivo?

-For western blot, primary and secondary antibodies details should be added (dilution, catalogue number, etc), also add molecular weight for bands detected

-The study will benefit from microscopic visualization of tumor sections. Also, immunostaining for HSPB2 and HSPB3 in tumor tissue can be performed

-More information regarding polymerase chain reaction should be added (cycling temperature and length)

Reviewer 3 Report

1, Please add the statistic results of Figure 1C. And discuss the change of protein HSPs expression level in different cell lines.

2, In this study, authors only analyzed mRNA level in Figure 2, 3, 6. Please add the results of protein levels (western blot and statistic results). 

3, Add the catalog number of chemicals, antibodies, and test kits.

Round 2

Reviewer 1 Report

The authors adequately responded to all comments and performed the required modifications

Reviewer 2 Report

The manuscript has been improved

Reviewer 3 Report

None.